# Cystathionine Gamma-Lyase Regulate Psilocybin Biosynthesis in *Gymnopilus dilepis* Mushroom via Amino Acid Metabolism Pathways

**DOI:** 10.3390/jof8080870

**Published:** 2022-08-18

**Authors:** Sen Yao, Chuanzheng Wei, Hui Lin, Peng Zhang, Yuanyuan Liu, Youjin Deng, Qianhui Huang, Baogui Xie

**Affiliations:** 1Mycological Research Center, College of Life Science, Fujian Agriculture and Forestry University, Fuzhou 350002, China; 2College of Life Science, Ningde Normal University, Ningde 352100, China

**Keywords:** CTH, L-cysteine, L-serine, psilocybin, UPLC, transcriptome, gene expression

## Abstract

As a potential medicine for the treatment of depression, psilocybin has gradually attracted attention. To elucidate the molecular mechanism regulating psilocybin synthesis in *Gymnopilus dilepis*, ultra-performance liquid chromatography (UPLC) was used to detect the changes in psilocybin content after S-adenosyl-l-homocysteine (SAH) treatment and the changes of psilocybin content in different parts (stipe and pileus), and RNA-Seq was used to explore the mechanism of psilocybin content changes. In this study, the psilocybin content in *G. dilepis* mycelia treated with SAH was significantly lower than that in the control group, and the content of psilocybin in the stipe was significantly higher than that in the pileus. Transcriptome analysis revealed that differential expression genes (DEGs) were associated with cysteine and methionine metabolism. In particular, the transcription levels of genes encoding Cystathionine gamma-lyase (CTH) in different treatments and different parts were positively correlated with psilocybin content. In addition, we found that the exogenous addition of CTH activity inhibitor (DL-propargylglycine, PAG) could reduce the content of psilocybin and L-serine, and the content of psilocybin and L-serine returned to normal levels after L-cysteine supplementation, suggesting that psilocybin synthesis may be positively correlated with L-cysteine or CTH, and L-cysteine regulates the synthesis of psilocybin by affecting L-serine and 4-hydroxy-L-tryptophan. In conclusion, this study revealed a new molecular mechanism that affects psilocybin biosynthesis, which can provide a theoretical basis for improving psilocybin synthesis and the possibility for the development of biomedicine.

## 1. Introduction

Psilocybin, the main alkaloid of fungi, is the primary component of hallucinogenic mushrooms. The molecular formula of psilocybin is C_12_H_17_N_2_O_4_P, and the molecular mass is 284.27. Additionally, psilocybin can be dephosphorylated to psilocin by phosphatases [1]. It was separated and identified firstly by Hofmann in 1958 [2]. As early as 1960, Sandoz sold tablets containing 2 mg of psilocybin as an auxiliary drug for the treatment of mental illness [3]. In recent years, more and more studies have demonstrated that psilocybin has potential medicinal value. Griffiths et al. [4] indicated that moderate amounts of psilocybin could produce a pleasurable experience through a 14-month follow-up study, and in conjecture, it may be caused by the hallucinogenic properties of psilocybin. Kometer et al. [5] found that psilocybin can enhance positive mood and proposed a possible antidepressant mechanism for psilocybin. Additionally, Grob et al. [6] firstly explored the potential ability of psilocybin to treat patients with anxiety disorders. Subsequently, psilocybin gained more and more attention in antidepressant research. Carhart-Harris et al. [7] found the depressive symptoms of patients were significantly reduced after psilocybin treatment, and psilocybin also relieves symptoms of anxiety and anhedonia. Davis et al. [8] suggested that psilocybin combined with psychotherapy was an efficient way of treating major depressive disorder (MDD) using a randomized clinical trial. Rootman et al. [9] showed that microdosing of psilocybin led to greater improvement in mood and mental health problems within a month compared to non-microdosing controls. In addition, hallucinogens have been shown to be a long-term effective addiction treatment [10], and there are several pieces of evidence suggesting that psilocybin has a clinically relevant role in the treatment of addiction. For instance, Johnson et al. [11] showed that psilocybin could improve the success rate of treating smoking addiction. Bogenschutz et al. [12] reported the effects of psilocybin in alcohol-dependent participants and provided preliminary safety data. Consequently, psilocybin has the potential to be effective in the treatment of psychological distress and neurosis due to its good absorption by the human body, long duration of efficacy, and few side effects, and it has become a research hotspot [13,14,15].

As an emerging research hotspot, psilocybin has been found to have different psilocybin contents in different stages and different parts of fungi. A study on cultivated *P. cubensis* published by Gotvaldová et al. [16] showed that there was no tryptamine in basidiospores, and the content of tryptamine alkaloids in fruiting bodies was higher than in mycelium. In addition, the content of tryptamine alkaloids in the caps was higher than that in the stipes. This indicates that psilocybin, as a tryptamine alkaloid, has obvious differences in different stages and different parts. Demmler et al. [17] used Ultra-High Performance Liquid Chromatography (UHPLC) combined with mass to quantitatively analyze psilocybin, and the results showed that the highest content of psilocybin was carpophores, followed by primordia, emerge mycelium, and submerse mycelium. Moreover, Demmler also found that S-adenosyl-l-homocysteine (SAH) hydrolase (SahH) and adenosine kinase (AdoK) can promote psilocybin biosynthesis by removing accumulated SAH in vitro. This means that the mastery of the synthesis mechanism of psilocybin can improve the production of psilocybin, which is beneficial to the mining of medicinal value. For example, Kargbo et al. [18] used the direct phosphorylation of psilocin to improve psilocybin production. In addition, Adams et al. [19] combined multiple genetic optimization techniques with fermentation techniques to increase psilocybin content by 32-fold. Heterologous expression of psilocybin synthesis genes in the order of P450 monooxygenases (PsiH), tryptophan decarboxylase (PsiD), kinase (PsiK), and methyltransferase (PsiM) in the mold *Aspergillus nidulans* to increase psilocybin production [20].

In recent years, with the development of science and technology, more and more researchers have paid attention to the biosynthesis of psilocybin, which provides a theoretical basis for the development and utilization of psilocybin in the future. The first study on the complete synthesis pathway of psilocybin was reported by Fricke et al. [21]; they found PsiD, PsiK, and PsiM were expected for the biosynthesis of psilocybin with 4-hydroxy-l-tryptophan as substrate, and PsiH, PsiD, PsiK, and PsiM can produce psilocybin with L-tryptophan as a substrate. These enzymes have been confirmed by in vitro methods. Moreover, the family transporters (PsiT1 and PsiT2) and transcriptional regulator (PsiR) were also found in *Psilocybe cubensis* and *Psilocybe cyanescens*. However, Awan et al. [22] found more than one psilocybin biosynthesis cluster or psilocybin pathway; their conclusion is similar to the result of Boyce et al. [23]. Furthermore, a transcription factor reported by Fricke [21] was not part of the psilocybin biosynthesis cluster. Reynolds et al. [24] reported the discovery of a psilocybin gene cluster in the genomes of *Psilocybe cyanescens*, *Gymnopilus dilepis*, and *Panaeolus cyanescens*. The gene cluster contains PsiD, PsiM, PsiH, PsiK, and PsiT genes that were horizontally transferred between fungi, and this paper supports the previously reported results by Fricke [21]. Blei et al. [25] showed a new enzymatic pathway of psilocybin production from 4-hydroxyindole and L-serine by adding tryptophan synthase (TrpB). Torrens-Spence et al. [26] reported aromatic L-amino acid decarboxylases (AAADs) from *P. cubensis* (*Pcnc*AAAD). They also indicated that *Pcnc*AAAD could catalyze the decarboxylation of L-tryptophan, the first critical step in psilocybin biosynthesis, and that calcium can enhance *Pcnc*AAAD activity to enhance psilocybin production. Furthermore, McKernan et al. [27] used comparative genomics to compare the genomes of 81 hallucinogenic mushrooms with the *P.envy* reference genome and found that the genomes of *P. galindoi*, *P. tampanens,* and *P. azurescens* have very low rates of read mapping compared to the previously reported *P. cubensis* genome.

Nowadays, the most well-known representative mushrooms that contain psilocybin are the *Conocybe*, *Copelandia*, *Panaeolus*, *Gymnopilus*, *Inocybe*, *Pluteus*, and *Psilocybe* genera. In particular, *Gymnopilus dilepis* (Basidiomycotina, Agaricales) is found in tropical and sub-tropical regions [28,29]. The cap is dark brown with small black scales, and the gills are yellow. The color of the stipe surface is similar to that of the cap, and there is an inconspicuous collarium. In particular, *G. dilepis* has been reported to be capable of producing psilocybin [21,24,30]; hence it is of great research significance. As published in previous studies [22,24], psilocybin biosynthesis gene clusters (PsiM, PsiD, PsiK, and PsiH) were detected in the *G. dilepis* genome. However, there are few studies on *G. dilepis* for psilocybin biosynthesis and applications.

In the present study, we provide a comparison of psilocybin content in the mycelium of *G. dilepis* after SAH treatment and a comparison of psilocybin content between the stipe and pileus of *G. dilepis*. Ultra-performance liquid chromatography (UPLC) was used to explore changes in psilocybin content because of its low cost, time-saving nature and easy operation [31,32]. RNA-Seq exhibited an advantage in detecting low abundance transcripts, and there was a high correlation between the qPCR data used to validate transcript expression and the results of RNA-Seq [33,34]. Therefore, RNA-Seq was applied to explore genes affecting psilocybin biosynthesis in this work. Furthermore, the transcriptome results were validated using the exogenous addition of DL-propargylglycine (PAG) and L-Cysteine. This work is different from previous studies on the synthesis pathway of psilocybin, and it may provide a new direction for investigating psilocybin biosynthesis.

## 2. Materials and Methods

### 2.1. Fungal Strain and Sample Collection

A dikaryon strain of *G. dilepis* was provided by the Fujian Edible Fungi Germplasm Resource Collection Center of China, which was isolated from wild-type strains. The mycelia of *G. dilepis* were cultivated on Potato Dextrose Broth (PDB, 200 g potato extract mixed with 20 g glucose to 1000 mL) liquid medium at 25 °C at 150 rpm in a shaker for 5 days. Additionally, the samples were divided into control and SAH treatment groups. The SAH group samples were added to 1 mM SAH solution, and the samples in the control group were added to the same amount of sterile water as the SAH solution. Samples were collected by filtration after 24 h of treatment. A part of the samples was immediately frozen in liquid nitrogen, then stored at −80 °C for RNA sequencing, and others were blotted with filter paper and air dried for psilocybin extraction. Cultivation of *G. dilepis* was carried out using the method of cultivating wood rot fungi modified by Yan et al. [35] (50% cottonseed hulls, 28% sawdust, 20% wheat bran, 1% calcium sulphate dihydrate, 1% sucrose and the total moisture content of 60%). The mature fruiting bodies of *G. dilepis* were separated into pileus and stipes and stored at −80 °C for further analysis. There were three biological replicates for each group of samples.

### 2.2. Psilocybin Extraction and Determination

The dried sample (0.2 g) was ground into powder and mixed with 4 mL methanol. The mixture was soaked overnight and ultrasound-extracted for 30 min. Additionally, the mixture was centrifuged at 5000 rpm for 10 min, and the supernatant was passed through a 0.45 μm filter membrane for testing [36].

The UPLC coupled with a photo-diode array (PdA) detector was used for the determination of psilocybin content on a Shimadzu LC-20A system (Shimadzu, Kyoto, Japan). The samples were separated in an Agilent HC-C18 chromatographic column (Agilent, Palo Alto, CA, USA) with a particle size of 5 μm, 250 mm × 4.6 mm inner diameter, at a column temperature of 30 °C and a flow rate of 0.5 mL/min. The detection wavelength was 220 nm. The injection volume was 4 μL. Mobile phase A was the mixture of 100 mM NaCl and 50 mM KH_2_PO_4_ adjusted to pH = 3 with H_3_PO_4_, and mobile phase B was methanol. The mobile phases were filtered on 0.45 μm membrane filters under vacuum and degassed by sonication for 30 min before use. The binary solvent-delivery gradient elution program was 3% B at 0.0–5.0 min; 3–10% B at 5.0–10.0 min; 10% B at 10.0–18.0 min; 10–3% B at 18.0–22.0 min; 3% B at 22.0–35.0 min [31]. The retention time for psilocybin was 18.7 min. The standard solution of psilocybin (1 mg/mL) was diluted with methanol to 0.625, 1.25, 5, 10, 20, and 40 μg/mL, and the linearity equations for psilocybin concentration and analyte area were plotted.

### 2.3. L-Serine Extraction and Determination

After optimizing the previous method [37], a 0.2 g sample was ground into powder and mixed with 800 μL deionized water according to the derivatization principle of phenyl isothiocyanate. The mixture was ultrasound-extracted for 2 h. Additionally, the mixture was centrifuged at 12,000 rpm for 10 min. Then, 400 μL of the supernatant were added to 200 μL of 0.1 mol/L phenyl isothiocyanate and 200 μL of 1 mol/L triethylamine acetonitrile solution, standing for 30 min at room temperature. The 800 μL n-hexane was added for extraction, repeated twice, and then sucked into the lower layer solution to be tested.

The L-serine detection equipment was the same as that used for psilocybin detection. The column temperature was 35 °C, and the flow rate was 1 mL/min. The detection wavelength was 254 nm. The injection volume was 5 μL. Mobile phase A was a mixture of 0.1 mol/L CH_3_COONa and acetonitrile (the volume ratio is 925 to 70) adjusted to pH = 6.5 with CH_3_COOH, and mobile phase B was methanol:acetonitrile:deionized water (20:60:20). The mobile phases were filtered on 0.45 μm membrane filters under vacuum and degassed by sonication for 30 min before use. The binary solvent-delivery gradient elution program was 0–2% B at 0.0–5.0 min; 2–5% B at 5.0–6.0 min; 5–9% B at 6.0–14.0 min; 9–21% B at 14.0–18.0 min; 21–45% B at 18.0–32.0 min; 45–55% B at 32.0–34.0 min; 55–100% B at 34.0–38.0 min; 100% B at 38.0–42.0 min; 100–0% B at 42.0–45.0 min; 0% B at 45–50 min. The retention time for the L-serine derivative was 10.4 min. The standard solution was diluted with deionized water to 1000, 500, 250, 100, 50, 10, and 0 μg/mL, and the linearity equations for L-serine derivative concentration and analyte area were plotted.

### 2.4. Transcriptome Analysis

An E.Z.N.A^®^ plant RNA kit (Omega, Stamford, CT, USA) was used for the isolation of total RNA. Three biological replicates were used for each treatment. The total RNA was sent to Tianjin Novogene Bioinformatics Co., Ltd. (Tianjin, China). For evaluation of data, the concentration and integrity of RNA were assessed, and the absorbance (OD value) of each sample was measured at the rate of 260/280 nm. For mRNA sequencing, a total amount of 1 µg RNA was extracted from each sample.

Sequencing libraries were generated using NEBNext^®^ UltraTM RNA Library Prep Kit for Illumina^®^ (New England Biolabs, Newburyport, MA, USA), and index codes were added to attribute sequences to each sample. The library preparations were sequenced on an Illumina Novaseq 6000 platform, and 150 bp paired-end reads were generated.

Fragments per kilobase per million mapped fragments (FPKM) were used to estimate the expression level of each gene. The relationship of the samples was analyzed using the Pearson correlation coefficient. The values of paired *t*-test < 0.05 and|log2 (fold change)| > 1 were calculated for identification of differential expression genes (DEGs). KEGG annotation was conducted based on the KEGG Automatic Annotation Server (KAAS) [38], and GO annotation was analyzed by the Blast2GO [39] and eggnog-mapper (Available online: http://eggnog-mapper.embl.de/, accessed on 23 June 2022). Omics Share (Available online: http://www.omicshare.com, accessed on 23 June 2022) and BMK Cloud (Available online: http://www.biocloud.net, accessed on 6 July 2022) were used for KEGG and GO enrichment analysis of DEGs.

### 2.5. Quantitative Real-Time PCR (qRT-PCR) Analysis

The cDNA was synthesized using total RNA by the RevertAid First-Strand cDNA Synthesis kit (TransGen Biotech, Beijing, China). The qRT-PCR reactions were analyzed with PerfectStart Uni RT-qPCR Kit (TransGen Biotech, Beijing, China). The housekeeping gene of Glyceraldehyde-3-dehydrogenase (gapdh) and the internal control gene of Ras-related small GTPase (ras) were used as an internal reference [40], and relative gene expression levels were calculated using the 2^−ΔΔCt^ method [41]. The qRT-PCR analysis of each sample was performed in three biological replicates.

### 2.6. Exogenous Addition Experiment

PAG is an effective inhibitor of cystathionine gamma-lyase (CTH). L-cysteine is a synthetic product of CTH. Therefore, 1 mM PAG and 0.1% L-cysteine were applied for validation of the CTH function. After cultivating the mycelia of *G. dilepis* at 25 °C, 150 rpm for 5 days, samples were assigned to three experimental groups, each group consisting of three biological replicates as follows:PAG: 1 mM PAG was added to the culture medium, followed by incubation for 24 h;PAG + L-Cys: 1 mM PAG and 0.1% L-cysteine were added to the culture medium, followed by incubation for 24 h;Control group: the same amount of sterile water as the other groups was added, and the sample was incubated for 24 h.

These samples were blotted with filter paper and air dried for psilocybin extraction.

### 2.7. Statistical Analysis

In the present study, the statistical significance was tested using a one-way analysis of variance (one-way ANOVA) or *t*-tests by SPSS 22.0 (SPSS Inc., Chicago, IL, USA). The estimated standard deviation (SD) was used to present the characteristics of the sample data, and the *p*-value < 0.05 was considered to determine whether the difference between the means was statistically significant.

## 3. Results

### 3.1. Comparative Analysis of Psilocybin in Samples from Different Treatments and Parts

UPLC was applied for the determination of psilocybin content. Firstly, the linear equation for the psilocybin standard was plotted. As seen in Appendix A, the peak time of psilocybin was 18.7 min; in the concentration range of 0.625 to 40 μg/mL, the linearity equation was y = 76,319x + 1306.6, and the correlation coefficient was R^2^ = 0.9997. The treated samples were extracted for psilocybin and detected by UPLC. The liquid chromatograms are shown in Appendix A. The absorption peak intensity of psilocybin in the control group was significantly higher than that in the SAH treatment group, and the absorption peak intensity of psilocybin in the stipe was also higher than that in the pileus. Furthermore, the psilocybin content of each sample was calculated according to the linearity equation. Figure 1a show the content of psilocybin in mycelial samples treated with SAH and without treatment. Figure 1b show the content of psilocybin in different parts of *G. dilepis* fruiting bodies. It can be seen from Figure 1 that the psilocybin content of the control group was almost twice that of the SAH treatment group, and the psilocybin content in the stipe was more than three times that in the pileus.

### 3.2. Sample Correlation Analysis and Differentially Expressed Genes (DEGs) Analysis

All samples were sequenced using Illumina paired-end sequencing. A total of 28.43 Gb of raw data and 27.18 Gb of clean data were obtained for the samples from different treatments, and a total of 26.74 Gb of raw data and 25.03 Gb of clean data were obtained for samples from different parts of the fruiting body. The transcriptome reads were compared with the existing genome, and the FPKM values were calculated for each gene. Pearson correlation analysis was performed on the FPKM values of these genes in each sample based on the BMK cloud platform. As shown in Figure 2a, the Pearson correlation coefficients of the SAH treatment samples were about 0.995–0.998, and the Pearson correlation coefficients of the control samples were about 0.937–0.993. This suggests that transcriptome data of different treatments were well correlated in biological replicate samples. Furthermore, the Pearson correlation coefficient of the SAH group and control group samples was not higher than 0.159. This result indicated that there were obvious differences among the different treated samples, and these differences may be the reason for the variation of psilocybin content. Meanwhile, Figure 2b exhibited that the Pearson correlation coefficient of pileus was about 0.929–0.962, and the Pearson correlation coefficient of stipes was about 0.973–0.992. It showed good correlation in biologically replicated samples for samples from different parts of the fruiting body. Furthermore, the Pearson correlation coefficient of samples in pileus and stipes was about 0.537–0.738. Although there is some correlation between the pileus and stipes, there are still obvious differences. In general, these samples have good repeatability within groups and obvious differences between groups, which can be used to study differential gene analysis affecting psilocybin syntheses.

In order to quickly and intuitively identify genes with large variation and statistically significant differences, the volcano plot of DEGs is exhibited in Figure 3. In the volcano plot, red dots represent significantly upregulated genes, green dots represent significantly downregulated genes, and black dots represent genes that are not significantly different. As shown in Figure 3a, compared with the control group, the SAH-treated samples have 1788 DEGs, of which 1159 DEGs were upregulated, and 629 DEGs were downregulated. The gene expression levels of the pileus and stipe samples were compared, and the results are shown in Figure 3b. There were 11,157 DEGs in the stipe samples, of which 7363 DEGs were downregulated and 3794 DEGs were upregulated.

### 3.3. GO Enrichment Analysis for DEGs

GO annotation of DEGs and enrichment analysis of annotation results. The analysis results of the SAH treatment group are shown in Figure 4a. There were 193 DEGs enriched into 31 GO categories, which were enriched and annotated into three main categories: biological process, cellular component, and molecular function. The GO analysis results of different parts of *G. dilepis* are exhibited in Figure 4b. There were 1913 DEGs enriched into 43 GO categories. There were many similarities between the results of the two GO enrichment analyses. For instance, in the biological process category, “cellular” and “metabolic process” were the major enrichment categories. This suggested that these DEGs were mainly involved in cellular composition and metabolic processes. In the cellular component category, “cell”, “cell part”, and “organelle” were the dominant categories, followed by “macromolecular complex” and “organelle part”. It means the cellular components in which these DEGs were located were mainly related to cells and organelles. In the molecular function category, “binding” and “catalytic activity” were the major enrichment categories. It showed that these DEGs might be closely related to enzymes. In this work, in order to shed more light on the synthesis of psilocybin, we paid more attention to DEGs related to “metabolic process” and “catalytic activity”.

### 3.4. KEGG Enrichment Analysis of the DEGs

Meanwhile, the Kyoto Encyclopedia of Genes and Genomes (KEGG) enrichment analysis was performed on the DEGs of different treatments and different parts of *G. dilepis*. A total of 138 DEGs in different treatments of mycelia were annotated, of which 89 upregulated DEGs in the SAH treatment group were enriched and 49 downregulated DEGs in the SAH treatment group were enriched (Appendix A). A total of 877 DEGs in different parts of the fruiting bodies were annotated, of which 293 upregulated DEGs in stipe were enriched, and 584 downregulated DEGs in stipe were enriched (Appendix A). Most of the DEGs were enriched in metabolic pathways, indicating that psilocybin changes may be related to multiple metabolic pathways, and this result was consistent with the GO enrichment analysis.

To characterize the DEGs most associated with psilocybin, the top 20 KEGG pathway enrichments for DEGs with different treatments and parts were exhibited in Figure 5 and Figure 6, respectively. In Figure 5, the upregulated genes in the SAH treatment group were significantly enriched in Tyrosine metabolism (ko00350) and Phenylalanine metabolism (ko00360). The downregulated genes were significantly enriched in the Selenocompound metabolism (ko00450) and cysteine and methionine metabolism (ko00270). The results indicate that amino acid metabolism may be related to the psilocybin synthesis pathway. In Figure 6, the top 20 KEGG pathway enrichments of upregulated DEGs in stipe were all significantly enriched, and most of them were related to amino acids such as the biosynthesis of amino acids (ko01230), arginine and proline metabolism (ko003300), alanine, aspartate, and glutamate metabolism (ko00250), cysteine and methionine metabolism (ko00270), arginine biosynthesis (ko00220), and glutathione metabolism (ko00480). However, the downregulated genes in stipe were enriched in cell cycle-yeast (ko04111), meiosis-yeast (ko04113), DNA replication (ko03030), mismatch repair (ko03440), homologous recombination (ko03440), and nucleotide excision repair (ko03420). This result may be caused by the growth characteristics of stipe and pileus of mushroom fruiting bodies. Previous studies [42] have shown that amino acids are one of the important pathways for alkaloid synthesis, so it is speculated that the amino acid metabolic pathways enriched in this part may be related to psilocybin synthesis.

### 3.5. Cysteine and Methionine Metabolism Is Correlated with Psilocybin Synthesis

In order to further mine the genes associated with changes in psilocybin content, the Venn diagram of upregulated and downregulated DEGs in different treatments and different parts of *G. dilepis* is exhibited in Figure 7. There are 69 DEGs in the overlapping sets of P_S_D and SAH_D, 42 DEGs in the overlapping sets of P_S_U and SAH_D, 54 DEGs in the overlapping sets of P_S_U and SAH_U, and 125 DEGs in the overlapping sets of P_S_D and SAH_U. Genes for which domains could be found in the 42 DEGs in the overlapping set of P_S_U and SAH_D and the 125 DEGs in the overlapping set are listed in Appendix A. In the overlapping sets of P_S_U and SAH_D combined with KEGG enrichment analysis, we found that the expression trend of the gene g6371 encoding CTH was positively correlated with psilocybin content, and the g6371 was enriched in the cysteine and methionine metabolism. Therefore, it is speculated that cysteine and methionine metabolism is correlated with psilocybin synthesis.

### 3.6. Validation of RNA-Seq Results Using qRT-PCR

The quantitative real-time polymerase chain reaction (qRT-PCR) was applied to validate the accuracy of RNA-Seq results. As mentioned above, psilocybin synthesis was related to cysteine and methionine metabolism. Therefore, qRT-PCR validation was performed using the DEGs enriched in cysteine and methionine metabolism. The g4167, g6371, and g6601 can encode CTH, so they were named *gdcth1*, *gdcth2,* and *gdcth3*, respectively. The g2558 can encode O-acetylhomoserine aminocarboxypropyltransferase (met17), so it was named *gdmet17*. The g4182 can encode serine dehydratase, so it was named *gdsds.* All primers used in this study are exhibited in Appendix A. As shown in Figure 8, the transcriptional levels of these genes were consistent with the transcriptome analysis results. Moreover, the expression levels of these genes have the same trend as psilocybin content. Therefore, this result indicated that the RNA-Seq data was accurate and reliable, and the DEGs related to cystine and methionine metabolism deserve further study.

### 3.7. CTH Related to Biosynthesis of Psilocybin

Next, we further confirmed that CTH is closely related to psilocybin synthesis. The gene and protein structure were drawn using Gene Structure Display server 2.0 (http://gsds.gao-lab.org/index.php, accessed on 6 July 2022) and Phyre 2 (http://www.sbg.bio.ic.ac.uk//phyre2, accessed on 6 July 2022), respectively. Phylogenetic tree analysis using MEGA 11 was also carried out. As shown in Figure 9a, the length of the *gd**cth2* was 1834 bp, including 10 exons and 9 introns. In Figure 9b, the protein structure of g6371 was predicted, and the domain was highly similar to cystathionine gamma-lyase. The phylogenetic tree analysis of several basidiomycetes, ascomycetes, plants, and animals’ CTH amino acid sequences is shown in Figure 9c. The samples were divided into four groups, and the *G. dilepis* was highly correlated with *Grifola frondosa* and *Sparassis crispa* of basidiomycetes. It suggested that the *gdcth2* is the gene encoding CTH.

As a potent CTH inhibitor, PAG was applied to verify the relationship between CTH and psilocybin biosynthesis. The result is shown in Figure 10a. The CTH enzyme activity of the PAG group was significantly lower than the control group. It indicated that PAG could act on CTH precisely and reduce the activity of the CTH enzyme. Meanwhile, psilocybin content was significantly reduced by the exogenous addition of PAG. L-cysteine is the catalytic product of CTH, so we further verified the relationship between CTH and psilocybin biosynthesis by adding L-Cysteine. The result is shown in Figure 10b. PAG treatment reduced psilocybin content, and the content of psilocybin increased significantly after supplementation with L-Cysteine. In summary, the result suggested that CTH (L-cysteine) was positively related to psilocybin biosynthesis.

In order to further investigate how CTH affects psilocybin synthesis, as shown in Figure 11a, we proposed the hypothesis that L-cysteine regulated psilocybin synthesis by affecting L-serine and 4-Hydroxy-L-tryptophan. In Figure 11b, the content of L-serine in the mycelium added with PAG was significantly lower than that in the control group. Additionally, the content of L-serine increased significantly after adding L-cysteine to the mycelium of the PAG group. The results prove that our hypothesis has certain credibility.

## 4. Discussion

As a potential antidepressant drug, several factors affecting psilocybin synthesis have been published. For example, the artificially synthesized mushroom kinase PsiK was applied to produce psilocybin from psilocin, indicating that the rational use of biocatalysis to increase the biosynthetic yield of psilocybin is feasible for medicinal use [43]. In addition, Kargbo et al. [18] also directly phosphorylated psilocin to obtain high-purity psilocybin for industrial production. However, the mechanism that affects psilocybin biosynthesis needs to be further studied and established. In this study, the transcriptome was used to study the expression of psilocybin synthesis-related genes. Transcriptome analysis revealed that 1159 DEGs were upregulated and 629 DEGs were downregulated between the control and SAH treatment groups. However, the published key psilocybin synthetic genes PsiM, PsiD, PsiK, and PsiH were not significantly different in transcriptome analysis after SAH treatment. Combined with the previous results, *Atheliaceae* detected close homologues of all psilocybin genes, but these genes were not clustered, and no psilocybin was detected [44]. Awan et al. [22] presented a novel psilocybin biosynthesis gene cluster in *Inocybe corydaline* that differs from the previous study. Furthermore, psilocybin was detected in cicadas parasitized by *Massospora* spp., although there was no typical synthetic gene cluster [23]. We inferred that a new metabolic pathway might be involved in the psilocybin synthesis pathway in *G. dilepis*. The speculation was similar to previous studies on plants; different species of plants produce serotonin and melanin in different biosynthetic pathways, and the enzymes involved have different substrate specificities [45].

We noticed that some DEGs in samples from different treatments and different parts were identified to be involved in the metabolic process and catalytic activity. In transcriptome analysis of different treatments of *G. dilepis*, the tyrosine metabolism, phenylalanine metabolism, and cysteine and methionine metabolism were significantly enriched. In transcriptome analysis of different parts of *G. dilepis*, KEGG pathway enrichment showed that DEGs were mainly involved in amino acid metabolism. For example, the alanine, aspartate, and glutamate metabolism was significantly enriched. As reported by Resende et al. [46], alanine-derived alkaloids were one of the main constituents of quinazolines. Additionally, tyrosine is one of the alkaloid precursors [47]. Moreover, amino acids are not only the nitrogen source but also the precursors of many compounds involved in plant reproduction, growth, and development [47,48,49,50]. Furthermore, the Hippo signaling pathway-multiple species has consistent roles in model animals, model plants, and model fungi, as well as affects cell proliferation and organ size by regulating cell mitotic exit [51,52]. Therefore, we inferred that psilocybin biosynthesis is correlated with growth and development. As reported by adding plant growth regulators to *Catharanthus roseus*, the results showed that the content of alkaloids increased significantly and was positively correlated with the concentration of growth regulators [53], which implied that cell growth and alkaloid content were closely related. Bienaimé et al. [54] found similar results in *Lycopodiella inundata*; the addition of growth regulators can strongly promote biomass growth and alkaloid production of the plant cell cultures. Therefore, transcriptomic data suggest that amino acid metabolism may be closely correlated with psilocybin content (alkaloid content).

In particular, DEGs with different treatments and DEGs in different parts have an identical gene *gdcth2*. The *gdcth2* encodes a CTH that catalyzes the production of L-cysteine. It has been revealed that psilocybin synthesis is related to cysteine and methionine metabolism. L-cysteine has been recognized as one of the key nutrient elements in the medium of tissue culture [55]. Furthermore, Effendi et al. [56] found that amnio acids were closely related to the cell growth of *Carica pubescens* callus, and different amino acids had different effects. Nurwahyuni et al. [57] also suggested that the addition of cysteine promoted the rapid growth of *Salacca sumatrana* callus and the formation of somatic embryos. Combined with the growth and alkaloid relationship described above [53,54], a bold hypothesis was proposed that psilocybin synthesis may also be related to L-cysteine or CTH, and L-cysteine regulates psilocybin synthesis by affecting L-serine and 4-hydroxy-L-tryptophan. In order to confirm the hypothesis mentioned above, the effects of inhibiting CTH activity (PAG) and supplementing L-cysteine (CTH product) on psilocybin biosynthesis were verified. In line with our prediction, the results indicated that inhibition of CTH activity could inhibit the synthesis of L-serine and psilocybin, and supplementation of L-cysteine can increase the content of L-serine and psilocybin to normal levels. In previous studies, Jeong et al. [58] suggested that redox-active cysteine residues were oxidized to thiosulfonates, which generated serine. Additionally, the serine produced 4-hydroxy-L-tryptophan with the action of L-tryptophan synthase (TrpB), which acted as a substrate to affect psilocybin synthesis [25]. Therefore, our hypothesis is theoretically feasible. However, few studies have shown that L-cysteine or CTH is involved in the alkaloid synthesis pathway. In future studies, it is essential to investigate the relationship between cysteine and psilocybin synthesis. In light of this evidence, one plausible explanation is that CTH may play an important role in psilocybin biosynthesis.

## 5. Conclusions

In summary, our results comprehensively revealed DEGs after SAH treatment and DEGs in different parts of *G. dilepis* using transcriptome technology. Furthermore, these DEGs were significantly enriched in GO terms and KEGG pathways. We found that psilocybin may be closely related to amino acid synthesis. In addition, we established that CTH is positively correlated with psilocybin synthesis. We further put forward a hypothesis that psilocybin synthesis may also be related to L-cysteine or CTH, and L-cysteine regulates psilocybin synthesis by affecting L-serine and 4-hydroxy-L-tryptophan. This study provides new ideas for future research on the mechanism of psilocybin synthesis and provides a suitable target point for improving the production of psilocybin.

## Figures and Tables

**Figure 1 jof-08-00870-f001:**
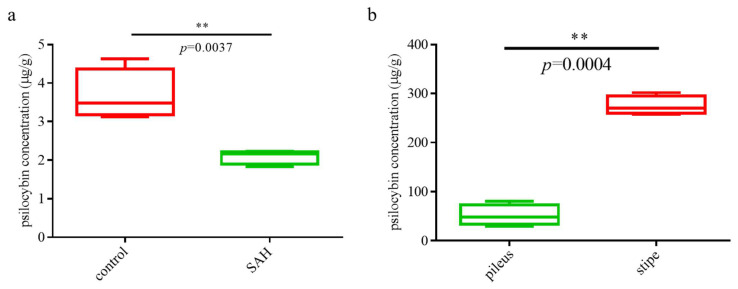
Quantitative analysis of psilocybin using UPLC ((**a**): Boxplots of psilocybin levels in samples after SAH treatment; (**b**): Boxplots of psilocybin levels in pileus and stipe; Paired *t*-test, ** indicated *p* < 0.01).

**Figure 2 jof-08-00870-f002:**
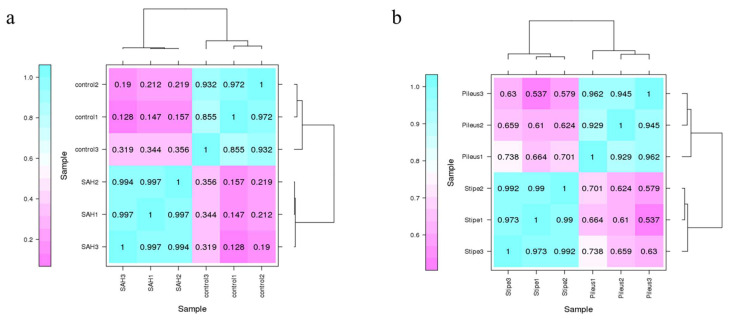
The Pearson correlation coefficients for relationships of transcriptomes data for different treatments (**a**) and different parts of the fruiting body (**b**).

**Figure 3 jof-08-00870-f003:**
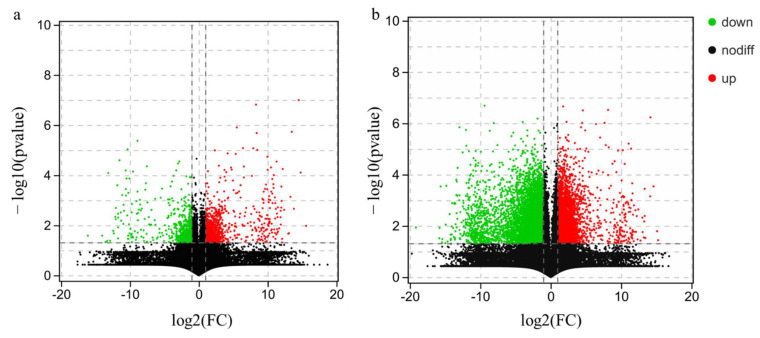
The volcano map of differentially expressed genes for different treatments (**a**) and different parts of the fruiting body (**b**).

**Figure 4 jof-08-00870-f004:**
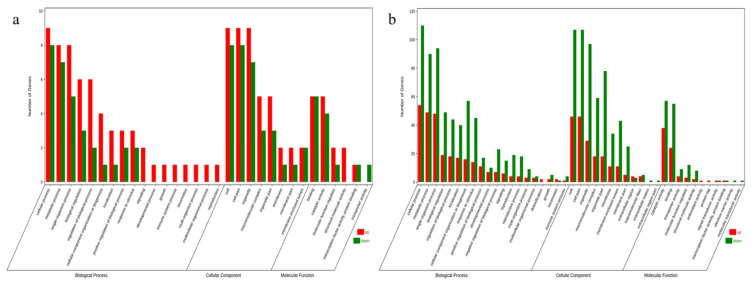
GO class of DEGs under different treatments (**a**) and different parts (**b**) of *G. dilepis*.

**Figure 5 jof-08-00870-f005:**
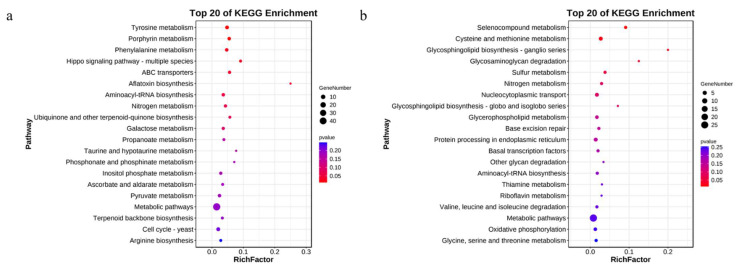
KEGG pathway enrichment analysis of DEGs under different treatments of *G. dilepis*. ((**a**): the upregulated genes; (**b**): the downregulated genes).

**Figure 6 jof-08-00870-f006:**
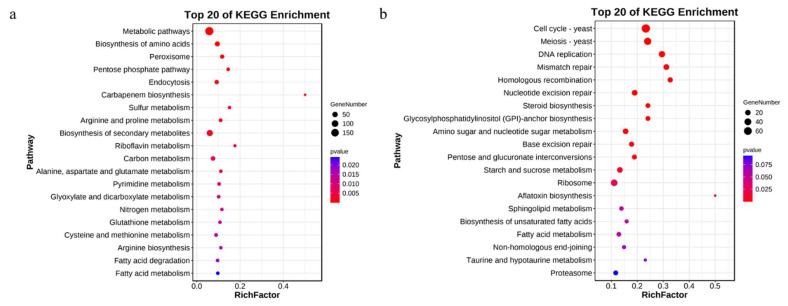
KEGG pathway enrichment analysis of DEGs under different parts of *G. dilepis*. (**a**): the upregulated genes; (**b**): the downregulated genes.

**Figure 7 jof-08-00870-f007:**
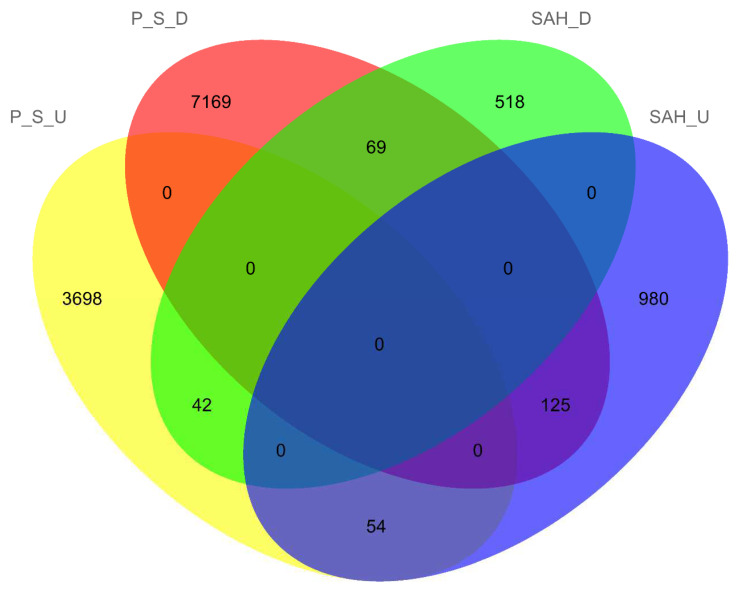
The Venn diagram of upregulated and downregulated DEGs in different treatments and different parts of *G. dilepis*. (P_S_U: upregulated DEGs in stipe compared with the pileus, P_S_D: downregulated DEGs in stipe compared with the pileus, SAH_U: upregulated DEGs in SAH group compared with the control group, SAH_D: downregulated DEGs in SAH group compared with the control group).

**Figure 8 jof-08-00870-f008:**
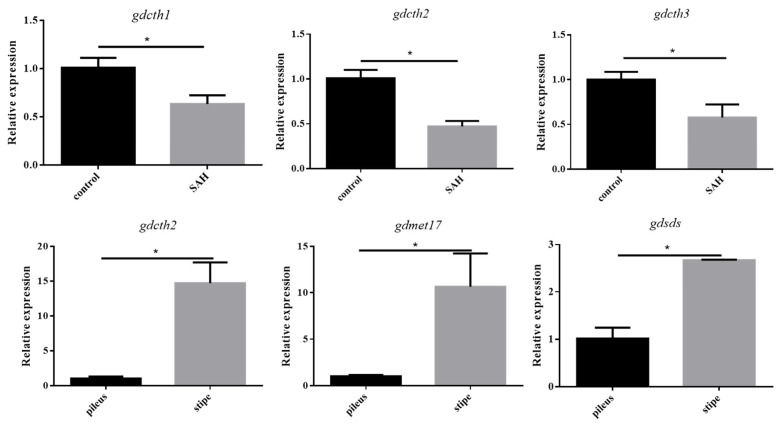
Validation of transcript expression changes by qRT-qPCR (Paired *t*-test, * indicated *p* < 0.05).

**Figure 9 jof-08-00870-f009:**
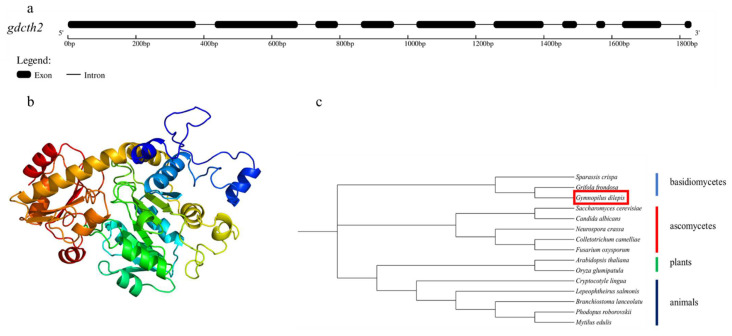
The characteristic of *gdcth2*. ((**a**): gene structure of *gdcth2*; (**b**): protein structure of CTH; (**c**): phylogenetic tree analysis of several CTH amino acid sequences).

**Figure 10 jof-08-00870-f010:**
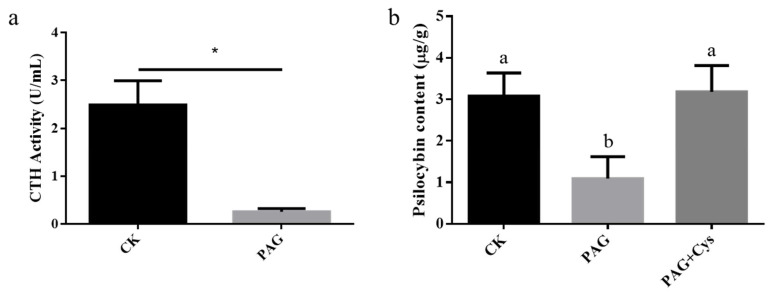
The impact of PAG and L-cysteine on psilocybin biosynthesis. ((**a**): The impact of PAG on CTH activity; (**b**): The impact of PAG on psilocybin content. Paired *t*-test, * indicated *p* < 0.05; different letters indicated significant differences).

**Figure 11 jof-08-00870-f011:**
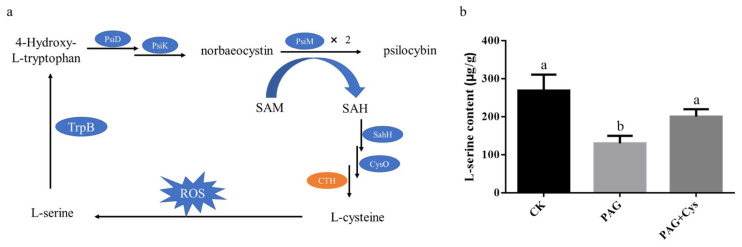
A schematic diagram of the hypothesis that CTH participates in the synthesis of psilocybin (**a**) and the effect of CTH on L-serine content (**b**). Bars with different letters (a and b) indicate a significant difference at *p* < 0.05.

## Data Availability

Not applicable.

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
