# Peer review of "Cystathionine Gamma-Lyase Regulate Psilocybin Biosynthesis in Gymnopilus dilepis Mushroom via Amino Acid Metabolism Pathways"

_jof, 2022, doi:10.3390/jof8080870_

Round 1

Reviewer 1 Report

The study in fact revealed a new molecular mechanism that affects the psilocybin biosynthesis.

- What can be the ultimate reason for the observation, that exogenous addition of CTH activity inhibitor (PAG) reduced the content of psilocybin and 21 L-serine?

- How could it be proved in an other way that psilocybin synthesis may be positively correlated with L-cysteine or CTH?

- How could you certify by means of different approaches, that psilocybin may be closely related to the amino acid synthesis?

Reviewer 2 Report

The current manuscript entitled “Cystathionine gamma-lyase Regulate Psilocybin Biosynthesis in Gymnopilus dilepis via Amino Acid Metabolism Pathways” by Yao et al. is a fascinating work that deep dives into the synthesis of the magic compound Psilocybin in Magenta Rustgill mushroom. After a careful reading, I found this manuscript interesting, timely, well-structured, and suitable for publication in JoF. The manuscript provides a well-formulated study on observation regarding the role of genes associated with CTH encoding during Psilocybin synthesis in different body parts of Magenta Rustgill mushroom. Overall, the study reveals that the molecular mechanism of Psilocybin synthesis can be effectively improved as a result of biotechnological modifications in CTH-encoding genes of the Magenta Rustgill mushroom. Considering the novelty of the work, I suggest accepting the manuscript for publication after minor revision. My specific recommendations are:

1.      Title: add “mushroom” after Gymnopilus dilepis.

2.      Abstract: Define all abbreviations used in this section, e.g., UPLC, SAH, etc.

3.      Avoid using keywords that already appeared in the title.

4.      The first sentence of the introduction has two times the use of the “main” word. Better to rephrase it.

5.      Add some biochemical characteristics (formula, mass, metabolism, etc.) of Psilocybin in the introduction section.

6.      Cite some relevant information from Paul Stamets who is on a magic mushroom mission for Psilocybin research.

7.      Line 61: UHPLC is not defined here, but later on 111?

8.      Line 124: Provide the composition of Potato Dextrose Broth in brackets.

9.      PDB and PDA are confusing, change PDA to PdA.

10.   Section 2.2: Reference for the Psilocybin extraction and analysis? Same for 2.3, 2.4, and 2.5 also.

11.   Check the accuracy of the degree sign (°).

12.   Section 2.6: Missing the number of samples and replicates.

13.   Figure 1 caption: add the abbreviation for CK. Same for other figures too.

14.   Figure 4: Axial text not visible. The figure can be expanded to the whole page i.e., horizontally.

15.   Somewhere authors used up and down while somewhere stipe and cap, be consistent in this regard.

16.   Figure 9: Add names of software used to draw 3D structure simulation. If adopted from the internet, add a source.

17.   Figure 11 caption and its discussion require strong referencing. It is just a proposed hypothesis that should be supported by relevant studies.

18.   Conclusion: well written.

Reviewer 3 Report

The current research article reports the topic that impacts the regulation of psilocybin biosynthesis in Gymnopilus dilepis. The authors elucidated the molecular mechanism of psilocybin biosynthesis in G. dilepis involves amino acid metabolism. The overall content of the manuscript is written in an average standard, the alignment of parts is almost appropriate. However, some of the improvement should be concerned. The metabolic pathway is too broad to identify the pathways that really involve the psilocybin synthesis. Specified method to detect specified intermediate compounds may be needed to elucidate the whole pathway. It is already known that amino acids involve psilocybin biosynthesis by other previous publications.

Abstract

The authors used abbreviations without the full names, reader who are not expertise in this field may do not understand. It would be better to use the full text, not abbreviation?

Line 13 suggest to put “S-adenosyl-l-homocysteine (SAH) treatment”

Line 16 suggest to use “in control group” instead of “in CK group”

Line 18 suggest to use “differential expression genes” instead of “DEGs”

Introduction

Line 58-59 The authors give example of different content of tryptamine in different part. However, the authors did not give information how tryptamine relates psilocybin? It would be better to give the information?

Line 71-71 The authors give example of heterologous expression of psilocybin, it is better to tell information of host and gene source?

Line 80-81 How to link this information to the psilocybin pathway? How this information is important? They were also found to involve in the synthesis?

Line 91 suggest to use “an aromatic L-amino acid” instead of “an Aromatic L-amino acid”

Line 76-80 Please recheck the content, there are only 3 enzymes (PsiD/PsiK/PsiM) reported by Fricke et al. that involve the synthesis from 4-hydroxy-l-tryptophan? PsiH involves the synthesis from L-tryptophan?

Material and methods

2.1 Sample collection

Line 128 suggest to put the methods to separate the mycelium sample e.g. “by filtration or centrifugation”?

Line 130 suggest to put drying method.

Line 130 The authors should give information for the reader how to get the mature fruiting bodies?

2.2 Psilocybin extraction and determination

Line 134 suggest “Dried sample (0.2 g) was ground into power and mixed with 4 mL methanol”

2.3 L-serine extraction and determination

Several “And then”, please try to rearrange the sentences.

Line 164 What the authors mean for “acetonitrile (925:70)”. The reader may be a bit confuse what the ratio it is. Please try to rearrange the sentence.

2.6. Exogenous Addition Experiment

Line 215 suggest the author put more detail about separating the mycelia from broth and dried the mycelia before use these samples for psilocybin analysis

The authors did not put   detail how to analyze psilocybin from stipe and pileus? The SAH in Figure 1 did not link to the methods. The authors should link in method’s abbreviation with the abbreviation in the result?

Result

Figure S1 suggest to use “arrow ” instead of “rectangle” because it makes the difficult for seeing the peak in chromatogram?

Line 226 suggest “ in the concentration range of ….”

Line 261, 269, 282 suggest “different parts of fruiting body” or “different parts of G. dilepis” like another Figure.

Line 334-335 suggest to put reference(s).

3.5. Cysteine and methionine metabolism is correlated with psilocybin synthesis

How to interpret the results between overlapping set gene, still have many genes involve and this might cause by other compound synthesis pathways?

3.6. Validation of RNA-Seq results using qRT-PCR

Line 363 suggest the authors to put gene name for each code in the manuscript, because the reader cannot fine the information in manuscript or supplementary data? Only code but expected gene name is not provided.

These genes are similar with any previous results by the other provious work?

The authors should explain why you are interested in the enriched in cysteine and methionine metabolism and try to follow these genes? Still lacks of the link from the genomic and metabolomic results?

Figure 8, 10 suggest to put p-value for * of comparison in the figure title. p-value versus? Should put all the gene results obtained from RT-PCR? Why the authors put only some genes in the results?

Table S3 suggest to put the reference(s)?

Should give the clear and related reason(s) why only gdcth2 are selected to study?

3.7. CTH Related to Biosynthesis of Psilocybin

The authors should provide the detail how gdcth2 and g6371 are related and why do you want to focus this.

Figure 9 suggest to combine the photo down to enlarge Figure 9a. This arrangement makes the difficult reading of the text in gene structure?

Figure 10 suggest to put CTH activity of all treatments to compare to the psilocybin content?

Any discussion why still being observed CTH activity and psilocybin, even through the enzyme inhibitor is used?

Figure 11 The authors said “L-cysteine reduces psilocybin synthesis” but no result to confirm this? because the authors did not put the psilocybin in the figure. The results of psilocybin should be compared with the concentration of cysteine?

Line 432-433 What the evidences that the authors inferred there may be new pathways for psilocybin synthesis in G. dilepis. I think the result is too broad to make this summary.

Round 2

Reviewer 3 Report

The current research article reports the topic that impacts the regulation of psilocybin biosynthesis in Gymnopilus dilepis. The authors elucidated the molecular mechanism of psilocybin biosynthesis in G. dilepis involves amino acid metabolism. The overall content of the manuscript is written in an average standard, the alignment of parts is improved and appropriate for next step.